# The Relationship between Physical Activity and Life Satisfaction among University Students in China: The Mediating Role of Self-Efficacy and Resilience

**DOI:** 10.3390/bs13110889

**Published:** 2023-10-27

**Authors:** Jiaxin Deng, Yongfeng Liu, Rui Chen, Yu Wang

**Affiliations:** 1School of Sports Training, Chengdu Sport University, Chengdu 610041, China; djx1999cdsu@163.com (J.D.); chen980324@163.com (R.C.); 2West China School of Basic Medical Sciences & Forensic Medicine, Sichuan University, Chengdu 610041, China; wangy0019@stu.scu.edu.cn

**Keywords:** mental health for university students, physical activity, self-efficacy, resilience, life satisfaction, chain-mediation role

## Abstract

(1) Background: Universities play a crucial role in nurturing healthy habits, and physical activity has emerged as a valuable tool for enhancing the life satisfaction, self-efficacy, and resilience of college students. Therefore, the purpose of this study was to investigate the mediating role of physical activity in the relationship between life satisfaction and self-efficacy as well as resilience among Chinese university students. (2) Method: This study used the Physical Activity Rating Scale (PARS-3), the Self-Efficacy Scale (CGES), the Mental Toughness Scale (CD-RISC), and the Life Satisfaction Scale (CSLSS) to administer questionnaires to 353 university students from two universities in Chengdu, Sichuan Province. Using a random simple sampling method, the data were processed and analyzed using SPSS 22.0 and Excel software. (3) Results: (1) Physical activity exhibited a significant positive correlation with self-efficacy, resilience, and life satisfaction; self-efficacy was significantly and positively related to resilience and life satisfaction; and resilience was significantly and positively related to life satisfaction. (2) Physical activity had a positive predictive effect on life satisfaction (β = 0.2771; 95% confidence interval (CI): 0.1905, 0.3637, 55.51%). (3) Self-efficacy (β = 0.0368; 95% confidence interval (CI): 0.0087, 0.0692, 7.37%) and resilience (β = 0.137; 95% confidence interval (CI): 0.0919, 0.1894, 27.44%) mediated the relationship between physical activity and life satisfaction. Also, the chain-mediating effect of self-efficacy and resilience between physical activity and life satisfaction reached significant levels (β = 0.0483; 95% confidence interval (CI): 0.0279, 0.0745, 9.68%). (4) Conclusion: Active participation in physical activity not only enhanced the life satisfaction of college students but also indirectly impacted their life satisfaction by improving their self-efficacy and resilience. Consequently, this led to an overall improvement in the life satisfaction of college students.

## 1. Introduction

The associations among life satisfaction, self-efficacy, mental resilience, and physical activity are highly relevant, even at the college level [1,2,3]. The importance of this type of research is supported by many negative indicators during this period, such as the increase in sedentary behavior among college students and a significant decrease in PA, which leads to lower quality of life, insomnia, and decreased life satisfaction, and is also associated with an increase in depression and other psychiatric disorders [4,5,6]. Globally, the mental health of college students has deteriorated [7,8,9], while the overall life satisfaction of college students is strongly associated with mental health risks [10]. Worryingly, under the dual pressures of employment and academic performance, psychological stress among college students is high [11], and educational pressures in schools contribute to the inability of students to participate in adequate recovery activities, especially adequate PA [12]. In addition, the gender differences within academic stress and employment stress increase with age [13], and, in most countries, college girls have higher academic stress than boys and increased stress overall [14].

There is no doubt that the college environment and college education have a significant impact on college students’ life satisfaction [15]. In addition to a good family background, school is an important component of the overall quality of life of university students [16]. In a study of Spanish university students, it was found that positive pro-social parity was significantly associated with life satisfaction and that self-esteem and positive daily activities were predictors of life satisfaction among university students [17].

It is well-known that participation in physical activity has a significant impact on increasing life satisfaction [18]. And, during the process, it has an effect on increasing self-efficacy and mental resilience [19,20]. In addition, it was shown that self-efficacy and mental resilience have a role in increasing life satisfaction [21].

Although there is an accumulation of research on the effects of physical activity on life satisfaction, some studies showed that physical activity also has a significant effect on self-efficacy and psychological resilience [22,23]. However, there is a lack of research on the effects of physical activity on life satisfaction among college students as a group, especially research that predicts the mediating role of self-efficacy and mental toughness in this process. Therefore, the purpose of this study was to investigate how physical activity can enhance college students’ life satisfaction from the perspective of positive psychology and to explore whether self-efficacy and mental toughness can mediate this relationship, to provide suggestions for improving college students’ mental health and promoting behavioral improvement.

## 2. Theoretical Foundations of Research

Physical activity involves engaging in activities of a certain intensity, frequency, and duration. It was found to effectively increase an individual’s self-confidence and positive emotional experiences. Additionally, there was a significant relationship between physical activity and self-efficacy, resilience, and life satisfaction [23,24,25]. Therefore, it is crucial to study the mechanisms through which physical activity affects self-efficacy, resilience, and life satisfaction among university students in order to effectively promote their physical and mental health development.

Currently, there is significant interest in understanding the relationship between physical activity and life satisfaction among university students. Previous studies suggested a positive association between physical activity and life satisfaction [26,27]. Furthermore, physical activity was found to have positive effects on mood, anxiety, and depression [28,29]. There was also evidence of both a direct and an indirect relationship [30] between physical activity and life satisfaction.

Self-efficacy is a crucial factor that influences an individual’s perceived ability to carry out a specific behavior. Previous research indicated that self-efficacy can help alleviate negative emotions and is an important predictor of life satisfaction in adolescents. While there are limited studies that specifically examine the predictive role of self-efficacy on life satisfaction, Moksnes found that self-efficacy has a significant impact on adolescent life satisfaction [31].

Furthermore, physical activity was shown to be an effective way to increase an individual’s self-efficacy [32], and there is a positive relationship between self-efficacy and physical activity [33]. Resilience is a positive personal attribute that refers to one’s capacity to overcome obstacles and restore a favorable state of life by relying on their own capabilities when faced with challenges, difficulties, and adversity. Resilience is believed to have a significant impact on increasing life satisfaction, as certain studies suggested a link between resilience and an enhanced sense of contentment in life [34].

Engaging in physical activity can effectively cultivate resilience, and there are notable differences in resilience levels among university students depending on their participation in sports [35]. The research showed a significant positive correlation between self-efficacy and resilience, indicating that an increase in self-efficacy leads to an increase in resilience [36]. Resilience acts as a protective mechanism for life satisfaction, while self-efficacy serves as a protective factor for resilience. Moreover, studies provided evidence for the predictive influence of physical activity on enhancing self-efficacy, resilience, and life satisfaction [37].

In summary, in order to explore the effects of physical activity behaviors on life satisfaction and the mediating role of self-efficacy and mental resilience in a group of college students, the following research hypotheses are proposed in conjunction with the theoretical and research bases mentioned above:

**H1.** 
*Physical activity significantly predicts life satisfaction among university students.*


**H2.** 
*Self-efficacy mediates the relationship between physical activity and life satisfaction.*


**H3.** 
*Resilience serves as a mediator between physical activity and life satisfaction.*


**H4.** 
*Self-efficacy and resilience have chain-mediated effects on physical activity’s influence on college students’ life satisfaction.*


## 3. Research Methodology

### 3.1. Study Population

Referring to the effect sizes of previous similar studies and considering the differences between the experimental design of the predecessor study and this paper, the sample size was reached with a minimum of 300 participants [38]. Power analyses for indirect paths indicated that the minimum sample size should be 300 [39]. A final total of 353 undergraduate students participated in this study. Therefore, an open-ended questionnaire was administered to 400 undergraduate students from two universities in Chengdu, Sichuan Province, China, through both in-person offline classroom communications and online distribution of the questionnaire using a simple random sampling method. We certify that this study was conducted in accordance with the 1964 Declaration of Helsinki and subsequent amendments. We confirm that all methods used in this study were approved by the ethics committee, and, in addition, we confirm that informed consent was obtained from all subjects and/or their legal guardians.

Written informed consent was obtained from all participants prior to inclusion in this study. The questionnaire was clearly explained to the subjects before distribution, and the questionnaire was independently completed with the subjects’ consent. After the completion of the overall survey, 47 questionnaires were screened and excluded for non-compliance or missing questions, resulting in 353 valid questionnaires, with a valid return rate of 88.25%. There were 146 males and 207 females in the sample; 178 (50.42%) of them were from rural households, and 175 (49.58%) were from urban households; and 84 were in the first year, 117 in the second year, 106 in the third year, and 46 in the fourth year, with an average age of 20.14 ± 2.31 years. Details of the participating university students are shown in Table 1.

### 3.2. Measurement Tools

#### 3.2.1. Physical Activity Rating Scale (PARS-3)

In this study, the Physical Activity Rating Scale, revised by Liang Deqing [40], was used to measure the level of physical activity of participants in terms of intensity, frequency, and duration of physical activity. The indicators were divided into five levels and rated on a scale of 1 to 5. Total score for physical activity = intensity of exercise × (duration of exercise-1) × frequency of exercise. The range is 0 to 100 points. The higher the score is, the higher the level of physical activity of the individual is. A score of ≤19 is considered a small amount of exercise; a score between 20 and 42 is a medium amount of exercise; and a score ≥43 is a large amount of exercise. In this study, the Cronbach’s α coefficient of the scale was 0.825.

#### 3.2.2. College Student Life Satisfaction Scale (CSLSS)

In this study, the CSLSS, revised by Wang Yuzhong [41], was divided into six objective satisfaction items (e.g., academic performance, relationship with friends, etc.) and six subjective satisfaction items (e.g., satisfaction with one’s life in general, etc.), using a five-point Likert scale with five levels from very satisfied to very dissatisfied. The fitted results of the measurement model were X^2^/df = 0.865, GFI = 0.993, CFI = 0.92, TLI = 0.95, RMSEA = 0.0001, and SSRMR = 0.017. The CSLSS Cronbach’s α coefficient was 0.904 overall.

#### 3.2.3. General Self-Efficacy Scale (GSES)

The General Self-Efficacy Scale (GSES), developed by Schwarzer [42], was used in this study. It has 10 questions on a 5-point Likert scale, ranging from very satisfied to very dissatisfied, with higher total scores indicating higher self-efficacy and vice versa. x^2^/df = 1.516, GFI = 0.970, CFI =0.992, TLI = 0.990, RMSEA = 0.038, and SRMR = 0.028, so the scale has good construct validity. The GSES‘s Cronbach’s α coefficient was 0.941 overall.

#### 3.2.4. Resilience Scale (CD-RISC)

The Chinese version of the resilience scale (CD-RISC), based on the Connor–Davidson Resilience Scale [43], used in this study, has three dimensions, namely, resilience, self-improvement, and optimism, with 25 items. Measurement model fit results were x^2^/df = 1.022, GFI = 0.936, CFI = 0.999, TLI = 0.999, RMSEA = 0.008. The overall Cronbach’s α coefficient for the CD-RISC was 0.943.

#### 3.2.5. Statistical Methods

The SPSS statistical package (version 22.0) was used for analysis. Categorical variables were expressed as frequencies and percentages. We used gender-adjusted analysis of covariance (ANCOVA), independent samples, and rank-sum tests, taking into account the results of Kolmogorov and normal distribution tests, to investigate the differences among physical activity, life satisfaction, resilience, and self-efficacy. Effect sizes, Cohen’s d or Pearson’s r, were calculated if *p*-values were obtained from parametric or non-parametric tests, respectively.

To test the possible mediating role of resilience and self-efficacy between physical activity and life satisfaction, we used model 6 of the SPSS Process 4.0 plug-in provided by Hayes for our analyses [44]. In this model, physical activity would be the independent variable (X variable), resilience and self-efficacy would be the mediating variables (M variable), and life satisfaction would be the dependent variable (Y variable). We also included age and grade as covariates in the study since they may influence life satisfaction. This analysis allowed us to estimate the indirect effects of resilience and self-efficacy in acting as mediators between physical activity and life satisfaction.

The analysis was conducted using SPSS, and the mediation hypothesis was tested using the bias-corrected bootstrap method on a sample of 5000, with confidence intervals (95%) calculated. Significance was considered when the confidence interval did not exceed zero.

## 4. Study Results

### 4.1. Common Method Bias

Possible common method bias was controlled for by anonymously filling out the questionnaire online, using positive and negative scoring, and the data was tested using the Harman one-way test, the KMO test, and Bartlett’s spherical test, which showed a KMO = 0.956, a Bartlett’s value of 10847.488, and a *df* = 1035, *p* < 0.001, indicating that the data were suitable for factor analysis. An exploratory factor analysis with all variables included as rotations extracted seven factors >1 with a maximum factor variance explained of 34.72% (<40%); therefore, the data for the study were not considered to be subject to a severe common method bias.

### 4.2. Correlation Analysis of Physical Activity, Life Satisfaction, Self-Efficacy, and Resilience

According to the findings in Table 2, physical activity was significantly and positively correlated with life satisfaction, self-efficacy, and resilience; life satisfaction was positively correlated with self-efficacy and resilience; and self-efficacy was positively correlated with resilience. In other words, there is a correlation between the above variables, in which self-efficacy and resilience may have certain influencing factors in the process of physical activity’s influence on life satisfaction, which provides a basis for the further calibration of mediating effects in the follow-up.

### 4.3. Multicollinearity Test

To test whether there is a problem of multicollinearity among the variables, in this study, the life satisfaction of university students was used as the dependent variable; physical activity, self-efficacy, and resilience were used as independent variables for covariance diagnosis; and the respective variables were standardized. The results showed that the tolerance values of the respective variables (0.760, 0.714, and 0.598) were greater than 0.1, and the variance inflation factor (VIF) values (1.315, 1.400, and 1.673) were less than 5. It can, therefore, be concluded that the data do not suffer from multicollinearity problems and are suitable for further chain-mediation-effect detection.

### 4.4. Chain-Mediated Effects Test for Self-Efficacy and Resilience

Bootstrap-mediated effects analysis was conducted using model 6 in Process 4.0, a SPSS macro program developed by Hayes. The model was specifically used to test for chain-mediated models using gender, race, age, and grade as control variables. The analysis was conducted using physical activity as the independent variable, self-efficacy and resilience as mediating variables, life satisfaction as the dependent variable, and gender and grade as control variables. In this study, the replicate sample was 5000, and the default confidence interval was 95% [45]. The results of the regression analysis (Table 3) and the data in Figure 1 show that physical activity was a positive predictor of life satisfaction (β = 0.267, *p* < 0.001) and had a significant positive prediction of self-efficacy (β = 0.280, *p* < 0.001) and resilience (β = 0.281, *p* < 0.001). Second, self-efficacy had a significant positive predictive effect on resilience (β = 0.354, *p* < 0.001) and life satisfaction (β = 0.126, *p* < 0.01). Third, resilience significantly predicted life satisfaction (β = 0.511, *p* < 0.001). This suggested that self-efficacy and resilience play a mediating role between physical activity and life satisfaction.

The results in Table 3 show that 44.49% of the total mediated effect of the mediation analysis and the 95% confidence interval (0.1285–0.3331) do not contain 0, indicating that the chain-mediation model with self-efficacy and resilience as mediating variables holds with some confidence. Secondly, by adding self-efficacy and resilience to the relationship between physical activity and life satisfaction, four pathways exist for the effect of physical activity on life satisfaction: Path 1: the direct effects of physical activity on life satisfaction are 0.2771, with an effect ratio of 55.51%; Path 2: the mediating effect of self-efficacy between physical activity and life satisfaction is 0.0368, with an effect ratio of 7.37%; Path 3: the mediating effect of resilience in physical activity on life satisfaction is 0.137, with an effect ratio of 27.44%; Path 4: the mediating effects of self-efficacy and resilience in the relationship between PA and life satisfaction are 0.0483, with an effect ratio of 9.68%. The 95% confidence interval for all four pathways is not zero, indicating that physical activity, self-efficacy, and resilience could independently influence life satisfaction, and self-efficacy and resilience could jointly play a chain-mediating role in the relationship between physical activity and the life satisfaction of university students, providing valid evidence for the above research hypothesis (Table 4).

## 5. Discussion

### 5.1. Relationship between Physical Activity and Life Satisfaction

The results of this study indicate that physical activity among university students has a positive influence on their life satisfaction. This finding confirms hypothesis H1 and is consistent with the findings of Liu et al.’s study, further supporting the correlation between physical activity and life satisfaction. Specifically, this study reveals that higher physical activity scores are associated with higher levels of life satisfaction among university students. This finding is of great clinical significance because low life satisfaction is known to be a key factor that leads to negative emotional states such as depression and even suicidal tendencies [46,47]. Physical activity can be used as an external form of relief to help people combat negative emotions, reduce depression, and decrease suicidal behavior in order to increase life satisfaction. Moreover, previous research showed that regular participation in physical activity can effectively satisfy the psychological needs and improve the subjective well-being of university students [48,49]. Furthermore, engaging in physical activity at a certain frequency and intensity can improve both the physical fitness and mental state of university students, promoting their overall physical and mental development [50]. This supports the hypothesis that physical activity has a significant predictive effect on life satisfaction among university students.

### 5.2. Mediating Role of Self-Efficacy in the Relationship between Physical Activity and Life Satisfaction

Based on the data in this study, it can be concluded that self-efficacy plays a mediating role in the positive relationship between physical activity and life satisfaction. The findings of this study provide support for Hypothesis 2. The results indicate that university students who do not engage in physical activity have weaker self-efficacy compared to those who do engage in physical activity. Additionally, individuals who do not engage in physical activity are more likely to experience negative psychological states such as depression, anxiety, and negativity. This can lead to a decrease in self-efficacy and an underestimation of one’s own abilities, ultimately resulting in lower life satisfaction. Furthermore, there is a positive correlation between self-efficacy and life satisfaction, indicating that self-efficacy can be a protective factor for life satisfaction. The research showed that self-efficacy is a positive personal resource that refers to an individual’s confidence in their own body, abilities, and emotions [51]. A high level of self-efficacy can result in a positive evaluation of oneself, leading to increased positive emotions and greater life satisfaction [52]. To sum up, the hypothesis that self-efficacy mediates the relationship between physical activity and life satisfaction is confirmed by the belief that university students can gain greater self-efficacy by engaging in physical activity, ultimately leading to higher levels of life satisfaction.

### 5.3. Mediating Role of Resilience in the Relationship between Physical Activity and Life Satisfaction

The concept of resilience has gained widespread recognition in positive and sports psychology research, with some studies suggesting it is a critical indicator of one’s mental health [53]. Previous research demonstrated that long-term engagement in physical activity can significantly enhance one’s level of resilience, contributing to improved subjective well-being [54]. A study conducted on university students examining the links among resilience, physical activity, and school adjustment found that physical activity is positively correlated with resilience and that there is a significant difference in resilience between physically active individuals and individuals who are not physically active [55]. Furthermore, resilience is considered a significant predictor of life satisfaction [56]. Past research suggested that resilience plays a crucial role in mediating the relationship between physical activity and life satisfaction. For example, a study on the effects of yoga participation on resilience and life satisfaction found that participants who engaged in weekly 60 min yoga sessions showed significant improvements in emotional well-being, resilience, reported life goals, and life satisfaction [57]. In this regard, a researcher [58] investigated the relationships among physical activity, resilience, self-efficacy, and life satisfaction among physically active university students and found that resilience mediated the relationship between physical activity and life satisfaction both independently and in conjunction with self-efficacy. These findings support Hypothesis 3, which suggests that resilience serves as a mediator between physical activity and life satisfaction.

### 5.4. Chain-Mediating Role of Self-Efficacy and Resilience in the Process of Physical Activity Affecting Life Satisfaction

This study showed that in the chain-mediated pathway between physical activity and life satisfaction, self-efficacy mediated the relationship between physical activity and resilience. Self-efficacy was significantly and positively correlated with resilience, which is consistent with Huang Shihua’s study [59]. This suggests that self-efficacy is an important psychological resource and has a positive effect on the formation and development of resilience. This study also showed that in the overall model, physical activity had a significant direct effect on psychological resilience, suggesting that physical activity has a positive effect on psychological resilience and acts as a protective factor for it, further supporting previous research findings.

Positive personality traits such as good communication skills, high self-efficacy, and strong emotion regulation were suggested as internal protective factors for resilience [60]. Internal and external protective factors play a key role in the formation and development of psychological resilience, promoting and stimulating its development in individuals [61]. Previous research showed that physical activity can enhance individuals’ self-efficacy, improve their communication skills, and enhance their emotional regulation [62]. Interestingly, all of these factors are elements of resilience. Therefore, physical activity and self-efficacy act as protective factors for resilience, and physical activity can indirectly influence resilience through the mediating role of self-efficacy.

In this study, resilience mediated the relationship between physical activity and life satisfaction, and resilience was significantly and positively associated with life satisfaction, which is consistent with Ramos-Diaz’s study. This suggested that resilience contributes to increased life satisfaction in individuals [63]. The root of this is that resilience is an important psychological resource for individuals, and an increase in resilience leads to more satisfaction, which, in turn, develops better life resources and equips individuals with greater emotional processing and the ability to suppress negative emotions, further increasing life satisfaction [64]. In other words, resilience facilitates life satisfaction, physical activity acts as a protective factor for resilience, and physical activity can indirectly influence life satisfaction through the mediating effect of resilience.

Overall, self-efficacy and resilience play a cascading mediating role in the influence of physical activity on life satisfaction. Among them, physical activity can indirectly affect life satisfaction through self-efficacy and also indirectly affect life satisfaction through resilience. Therefore, the chain-mediating effect was found to exist in the integration and expansion of the relationships among self-efficacy, resilience, and physical activity on life satisfaction, which contribute to further understanding the reasons for the influence of physical activity on college students’ life satisfaction. The study has a positive impact on enhancing individual self-efficacy and psychological resilience, thus improving individual life satisfaction and promoting the development, behavioral improvement, and physical and mental health of university students.

### 5.5. Research Limitations and Future Research Directions

There were also shortcomings of this study. Firstly, due to objective factors such as time constraints and limited research funding, this study adopted a cross-sectional research design. Although previous studies provided a solid foundation for this study, the findings of this study can be further expanded and enriched through follow-up studies and empirical research.

Secondly, the subjects of this study were all undergraduate university students, and there were no population differences considered. In a follow-up study, we can further discuss the effects of physical activity on life satisfaction in different age groups and racial backgrounds.

Finally, the physical activity and life satisfaction indicators in this study were self-reported by the participants through questionnaires. In the future, we hope to further explore the effects of the frequency, intensity, and type of physical activity on life satisfaction.

## 6. Conclusions

Physical activity was identified as a significant predictor of life satisfaction among university students. Self-efficacy and resilience played mediating roles in the relationships among personal development, physical activity, and life satisfaction among university students.

The findings of this study have implications for individual university students, the university, and society as a whole, with respect to participating in physical activity and enhancing life satisfaction. Firstly, it is crucial for university students to maintain a regular physical activity routine in order to improve life satisfaction and boost self-efficacy and enhance resilience. University institutions should prioritize students’ physical activity by providing the necessary facilities, resources, and conducive conditions for their participation in physical activities. Lastly, society should address the issues of educational and employment pressures faced by university students by implementing policies that not only reduce these pressures but also promote opportunities for students to engage in sports and enhance their life satisfaction. This holistic approach can help alleviate the burdens of education and employment and, ultimately, improve the well-being of university students.

## 7. Research Limitations and Future Research Directions

There were shortcomings of this study. Firstly, due to objective factors such as time constraints and limited research funding, this study adopted a cross-sectional research design. While previous studies provided a solid foundation, the findings of this study can be expanded and enriched through follow-up and empirical studies.

Second, the subjects of this study were all undergraduate college students, and population differences were not considered, especially regarding gender, age, grade levels, and ethnic backgrounds. In a follow-up study, we can explore differences related to physical activity among various groups.

Finally, both the physical activity and life satisfaction indicators in this study were self-reported by participants through questionnaires. In the future, we hope to further investigate the effects of the frequency, intensity, and type of physical activity on life satisfaction.

## Figures and Tables

**Figure 1 behavsci-13-00889-f001:**
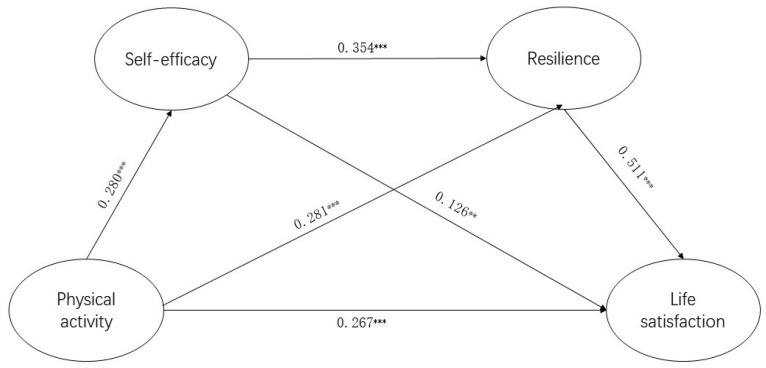
Chain-mediation model of self-efficacy and resilience. ** *p* < 0.01, *** *p* < 0.001.

**Table 1 behavsci-13-00889-t001:** Demographic variables of participants (N = 353).

Projects	Categories	Cases	Percentage
Gender	Male	146	41.40%
	Female	207	58.60%
Age	19	110	31.16%
	20	90	25.50%
	21	74	20.96%
	22	32	9.07%
	>22	47	13.31%
Grade	Freshman	84	23.80%
	Sophomore	117	33.14%
	Junior	106	30.03%
	Senior	46	13.03%
Residence	Urban	175	49.58%
	Rural	178	50.42%

**Table 2 behavsci-13-00889-t002:** Correlation analysis of physical activity, life satisfaction, self-efficacy, and heart resilience among university students (N = 353).

Variables	Mean (*M*)	Standard Deviation (*SD*)	1	2	3	4
Physical activity	49.220	36.932	1			
Life satisfaction	3.306	0.957	0.546 **	1		
Self-efficacy	3.395	0.936	0.325 **	0.424 **	1	
Resilience	3.372	0.769	0.524 **	0.602 **	0.534 **	1

** *p* < 0.01.

**Table 3 behavsci-13-00889-t003:** Regression analysis of chain-mediated models of physical activity, life satisfaction, self-efficacy, and resilience among university students.

Regression Equation		Overall Fit Index			Regression Coefficient				
Resulting variables	Predictor variables	R	R^2^	F	*β*	SE	t	LLCI	ULCI
Self-efficacy	Physical activity	0.3117	0.0972	12.4840 ***	0.280	0.0479	5.8442 ***	0.1857	0.3742
Resilience	Physical activity	0.6458	0.4171	62.0672 ***	0.281	0.0332	8.4853 ***	0.2161	0.3465
-	Self-efficacy	-	-	-	0.354	0.0354	9.9939 ***	0.2842	0.4234
Life satisfaction	Physical activity	0.6711	0.4504	56.7028 ***	0.267	0.0440	6.0750 ***	0.1807	0.3537
-	Self-efficacy	-	-	-	0.126	0.0485	2.5983 **	0.0306	0.2215
-	Resilience	-	-	-	0.511	0.0648	7.8755 ***	0.3830	0.6380

** *p* < 0.01, *** *p* < 0.001.

**Table 4 behavsci-13-00889-t004:** Chain-mediation test for self-efficacy, resilience on physical activity, and life satisfaction.

Effective Models	Effect Size	Boot SE	Bootstrap 95% CI		Proportion of Relative Effect
			Boot LICI	Boot ULCI	
Total effect	0.4992	0.0439	0.413	0.5855	100.00%
Physical activity → life satisfaction	0.2771	0.0441	0.1905	0.3637	55.51%
Physical activity → self-efficacy → life satisfaction	0.0368	0.0153	0.0087	0.0692	7.37%
Physical activity → resilience → life satisfaction	0.137	0.025	0.0919	0.1894	27.44%
Physical activity → self-efficacy → resilience → life satisfaction	0.0483	0.0118	0.0279	0.0745	9.68%
Total intermediary effect	0.2221	0.0521	0.1285	0.3331	44.49%

## Data Availability

The raw data supporting the conclusions of this article will be made available by the authors without undue reservation.

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
