# Peer review of "The Relationship between Physical Activity and Life Satisfaction among University Students in China: The Mediating Role of Self-Efficacy and Resilience"

_behavsci, 2023, doi:10.3390/bs13110889_

Round 1

Reviewer 1 Report

Comments and Suggestions for Authors

The study is interesting but needs significant changes to improve the overall quality. Specific points are given below;

1.      Abstract

Background: To analyze how physical activity influences the life satisfaction of university students and the relationship between those factors. It’s not a scientific way to write the background. The authors have mixed both background and objective of the study. It seems they don’t know the difference between the background and objective of the study. They have to redesign and rewrite the complete abstract.

   2.     Introduction

The introduction section is concise and lacks scientific facts about physical activity and life satisfaction. They must write a comprehensive and detailed introduction using the global-to-local approach.

3. I’m unable to find the research gap or statement of the study. They have to write a statement of the study after writing the introduction section.

4. The theoretical foundation of the research section also needs to improve. Don’t put your hypotheses in the paragraph. The hypothesis should be under the paragraph in a new line.

5.         Research Methodology

I would like to see the complete questionnaire (English translation). Provide the complete questionnaire in English with the revised manuscript. Provide the correlation matrix table of the study variables. Write statistical methods in the form of a single paragraph.

6. The conclusion section is very short and doesn’t reflect the main findings and crux of the study. The authors should rewrite the conclusion section

7.  What is the implication of your study? Write the three levels implications (Societal level, Individual level, and Organizational level)

8.  Write the limitations and future research directions after the conclusion section

Comments on the Quality of English Language

Extensive English editing is required.

Author Response

Thank you very much for your valuable comments on the manuscript. We have made changes based on your comments, please see the revised version of the manuscript and cover letter uploaded by us for specific changes and responses.

Reviewer 2 Report

Comments and Suggestions for Authors

The manuscript by Jiaxin D. et al. is well-written and well-described, focusing on the life satisfaction, physical activity, self-efficacy, and resilience of undergraduate students in Chengdu, Sichuan Province. Minor comments are suggested:

1. The abstract (Page 1 lines 21-22) mentions p-values from the results presented in Table 4. However, Table 4 does not have p-values. This needs to be corrected by adding p-values to Table 4 or replacing the p-values for the percent of proportion of relative effect. On this same note, below Table 4 are p-values presented they should be added to the Table or removed. In addition, the beta-value of resilience (line 22) mentions B=0.0137. However, in Table 4 the Effect size or B-value = 0.137. This needs to be corrected.

2. The percentage values on Page 3 lines 125 and 126 are interchanged. It should be corrected to 178 (50.42%) and 175 (49.58%).

3. Should the value of RMSEA (Page 4 line 147) be RMSEA = 0.000 or 0.0001?

4. The acronyms M and SD should be described in Table 2. Mean (M) and Standard Deviation (SD).

5. The acronym VIF on Page 5 line 199 should be described as Variance inflation factor (VIF).

6. The acronyms R, F, SE, t, LLCI, and ULCI should be described in Table 3.

7. Table 3 is missing the description of **** P-value.

8. More importantly, was there any difference comparing male and female, age groups, and college grade levels for life satisfaction, physical activity, resilience, and self-efficacy?

9. The manuscript is missing a section on author contributions.

Comments on the Quality of English Language

1. Page 3 line 112 is missing punctuation between "results of this study. Therefore,...".

2. Page 3 line 112 the word "An" should be in lowercase letters.

3. Page 4 line 154 the word "the" should be in uppercase letters.

4. Page 5 line 194 the word "In" should be in lowercase letters.

5. Page 5 line 210 remove punctuation between "Figure 1 show. physical...".

6. Add "life satisfaction" on Page 5 line 215.

Author Response

(The authors gave the same response as above.)

Reviewer 3 Report

Comments and Suggestions for Authors

First of all, I would like to thank the Editor-in-Chief and Guest Editors from Behavioral Sciences for the opportunity to prepare a review report for the Manuscript-ID: behavsci-2650175 entitled “The relationship between physical activity and life satisfaction among university students in China: the mediating role of self-efficacy and resilience”. According to the authors, the purpose of the present work under consideration for publication was twofold: i) to analyze how the physical activity influence life satisfaction of university students and the relationship between those factors and ii) examine the mediating effects of self-efficacy and Resilience, and to offer recommendations for enhancing the life satisfaction, positive emotions, and behavioral patterns of university students. Working hypothesis were that i) Physical activity significantly predicts life satisfaction among university students; ii) Self-efficacy mediates the relationship between physical activity and life satisfaction; iii) Resilience serves as a mediator between physical activity and life satisfaction and iv) self-efficacy and resilience have a chain mediating effect in the process of physical activity that influences the life satisfaction of university students. There were 146 males and 207 females in the sample. Main outcomes indicated that physical activity was a significant predictor of life satisfaction among university students. Self-efficacy and resilience could play a mediating role in both the individual development of university students and the influence of physical activity on life satisfaction. While I appreciate the efforts made by the authors to include various participants and the subject covered by the present study can indeed have impact on the population studied, there are various aspects lacking scientific robustness. As a consequence, before I can make a final decision on the acceptability of the present manuscript, there are MAJOR REVISIONS required:

P1L32. Across the introduction I strongly recommend avoid extensive literature reviews as it is not the aim of the current work

P1L62. Introduction is presented with various isolated paragraphs, please attempt to be much more to the point (specific) while avoiding excessive literature review. As a suggestions, focus on the study question by stating: 1) what exist, 2) what is the importance of the topic and 3) what is still necessary (the present work). No more than 3-4 paragraphs in length could improve the flow and readers interest.

P3L104. “Based on these findings, this study proposes hypothesis H4, suggesting that self-efficacy and resilience have a chain mediating effect in the process of physical activity that influences the life satisfaction of university students. Additionally, a hypothesis model is proposed to support this proposed effect.” - Insert hypothesis (and their support) together at end of introduction, immediately following study aim

P3L110. “Regarding the estimation of the sample size, we estimated the sample size in advance based on previous studies, and concluded that a sample of 300 people could potentially provide some validation of the results of this study Therefore, An open-ended questionnaire was administered to 400 undergraduate students from two universities in Chengdu, Sichuan Province, China, through both offline classroom communication in person and online distribution of the questionnaire using a simple random sampling method.” - It is necessary to report in full the procedure to estimate sample size as in the present form it is lacking important information. In addition to the protocol, please also cite “previous studies” accordingly

P4L160 “3.2.5. Statistical methods

Data were analyzed using SPSS 22.0, Bootstrap method, and SPSS macro plugin Process 4.0 plugin (Model 6), in order to perform questionnaire entry, statistical analysis, and 162

chain mediated effects test.  

Firstly, the reliability of the internal consistency of the variables was tested by Cronbach's alpha coefficient.

Secondly, the presence of common method bias in each variable (physical activity, self-efficacy, resilience and life satisfaction) was tested by Harman's one-way test.

Again, the mean, standard deviation and correlation coefficient of each variable were calculated using SPSS 22.0 software to diagnose covariance for possible multicollinearity.

Finally, regression analyses and the macro plugin Process 4.0 were used to test for independent and chained mediating effects on physical activity and life satisfaction.” - There are various isolated paragraphs which does not make sense. Please revise this whole subsection and provide additional pertinent information e.g. p-value and correlation thresholds, normality of data distribution, effect sizes

P5L183. “According to the findings in Table 2, physical activity was significantly and positively correlated with life satisfaction, self-efficacy and resilience; life satisfaction was positively 1correlated with self-efficacy and resilience; and self-efficacy was positively correlated with resilience.”- insert qualitative judgements on the correlation magnitudes to improve meaningful of results

P5L208. “In this study, the replicate sample was 5000 and the default confidence interval was 95%.” - Why  these values were used? reference(s) needed

P6L226. “The direct effect of path 1 (physical activity → life 226 satisfaction) is 0.2771, with an effect ratio of 55.51%; the mediating effect of path 2 (physical activity → self-efficacy → life satisfaction) is 0.0368, with an effect ratio of 7.37%; the mediating effect of path 3 (physical activity → resilience → life satisfaction) is 0.137, with an effect ratio of 27.44%; The mediating effect of path 4 (physical activity → self-efficacy → resilience → life satisfaction) was 0.0483, with an effect ratio of 9.68%.” - I strongly recommend avoid using “→” in the text while it could be preferably used in figures/tables only

Author Response

(The authors gave the same response as above.)

Reviewer 4 Report

Comments and Suggestions for Authors

Thank you for your work in the manuscript.

1. I’ve identified several grammatical errors and typos. Please have the manuscript proofread by a copyeditor.

2. Introduction ends quite abstractly, missing what has been known in previous research, what has not been known, and how the manuscript can fill the gap. These should be addressed as well as a clear research purpose statement.

3. Hypotheses statements in the section 2 relied heavily on previous research findings. I’m not saying this is wrong, but it doesn’t tell us “why” hypotheses would work. In addition to citing exiting research findings, the authors should clearly explain why A would influences B.

4. I’d suggest creating a “practical implication” section where the authors provide some suggestions to practitioners who can improve college students’ well-being. The authors should tell us how the findings can help improve college students well-being.

Comments on the Quality of English Language

Readability needs to be improved.

Author Response

(The authors gave the same response as above.)

Round 2

Reviewer 1 Report

Comments and Suggestions for Authors

NA

Comments on the Quality of English Language

English editing is required

Author Response

Many thanks to the reviewers for their work, and we take your suggestions for English editing very seriously. We have invited our colleagues who are proficient in English to conduct a thorough grammar check and English expression enhancement of our manuscript. Please see the uploaded revised manuscript for the specific changes. Thank you again for reviewing our manuscript.

Reviewer 3 Report

Comments and Suggestions for Authors

Thank you for addressing my suggestions

Comments on the Quality of English Language

The authors did not mention whether the article was revised concerning the English by a native speaker

Author Response

Thank you very much for the reviewer's work. We attach great importance to your suggestion. The scale that can be improved has been based on your suggestions and the full text of the further revision. We believe that with your suggestions and our work efforts, the quality of the manuscript has been effectively improved. For the specific part of the revision, please check the revised version of the manuscript we submitted. Thank you again for your suggestions.

Reviewer 4 Report

Comments and Suggestions for Authors

Thank you very much for addressing my comments.

Author Response

Many thanks to the reviewers. We are confident that with your suggestions and our work efforts, the quality of the manuscript has been effectively improved. Once again, we thank you for your suggestions.
